# Why Are You Running and Does It Hurt? Pain, Motivations and Beliefs about Injury Prevention among Participants of a Large-Scale Public Running Event

**DOI:** 10.3390/ijerph16193766

**Published:** 2019-10-07

**Authors:** Jan Wilke, Oliver Vogel, Lutz Vogt

**Affiliations:** Department of Sports Medicine, Frankfurt am Main, Goethe University Frankfurt am Main, 60488 Frankfurt am Main, Germany; vogel@sport.uni-frankfurt.de (O.V.); l.vogt@sport.uni-frankfurt.de (L.V.)

**Keywords:** pain, running, race, physical activity, exercise, musculoskeletal

## Abstract

Organized running events have gained substantial popularity. This study aimed to elucidate the prevalence of musculoskeletal pain, knowledge about injury prevention as well as the attitudes and motivations of individuals participating in the JP Morgan Corporate Challenge in Frankfurt (Germany). A total of 720 recreational runners completed a digital questionnaire immediately prior to the start. The majority of them displayed low to moderate physical activity levels and were rather unambitious regarding targeted finishing time. One quarter (25.3%) participated for the first time in an organized race. The most stated reasons to register were team building (76.4%) and experiencing the run’s atmosphere (50.6%). In contrast, improving health played a minor role (19.4%). More than one in five individuals (*n* = 159 runners) reported pain, with the most common locations being the knee and lower back. Both at rest (3.2/10 on a numerical rating scale) and during activity (4.7/10), average pain intensity was clinically relevant. Almost three thirds of the participants believed that stretching and wearing appropriate shoes would be effective for injury prevention while other methods such as resistance training, balance exercise or wearing of orthoses were rarely named. Musculoskeletal pain is a significant burden in runners participating in an urban mass event. In view of the poor knowledge about injury prevention, organizers and coaches may consider offering structured preparation programs as well as tailored running-related health education.

## 1. Introduction

Regular engagement in physical activity has been shown to represent a powerful method to prevent modern society diseases and reduce all-cause mortality [1,2]. Notwithstanding, although there is a large body of evidence supporting the health-beneficial effects of movement [1,2], the majority of the world population fails to meet international guidelines detailing tailored advice for a variety of populations [3]. Sustainably motivating sedentary individuals to create and maintain an active lifestyle may have a significant impact from a public health perspective: according to calculations of Lee et al. [4], inactivity causes more than 5.3 million deaths per year and its elimination would increase the worldwide life expectancy by 0.68 years.

Health associations and policy makers have used a plethora of approaches to promote regular physical activity. In addition to media campaigns, active public transport (e.g., biking) or the creation of a movement-friendly environment [5], the organization of mass sports events may foster the development of a more active society [6,7]. In recent decades, recreational running has experienced a dramatic surge in popularity. For instance, in Finland, the number of running events has tripled between 1979 and 2010 and in Greece, the number of event participants increased from 13.576 to 68.999 between the years 2006 and 2012 [8]. 

Despite the running boom in western countries, the prevalence of pain during mass events, the ‘typical’ participant of public mass events, his motivations to participate as well as his knowledge towards effective injury prevention are largely unknown. Most available trials examining recreational running have focused on the effect of preventive interventions or the prospective registration of injuries [9,10,11,12]. However, although counting health problems over a longer time frame is indispensable for the estimation of injury risk, the prevalence of discomfort immediately prior to a specific event represents another important factor. Firstly, acute pain may modify running biomechanics creating overload or, if becoming more severe, cause a higher degree of structural damage than before. Secondly, when seen from a public health perspective, pain may affect self-efficacy of running novices, decreasing their desire to further engage in physical activity. 

To date, only one study has examined pre-race pain in recreational runners [13]. Twenty-two percent of the 1049 surveyed participants covering distances between 5000 and 10,000 meters reported discomfort before the start of the race. While this finding yields indications for a considerable share of runners in pain, it should be noted that the study authors cumulated the data from five different races. Similarly, only two surveys [14,15] investigated the beliefs of recreational runners of effective methods for injury prevention. Van Mechelen et al. [14] obtained questionnaires assessing the knowledge on popular strategies such as stretching, warm-up and cool-down in Dutch participants. However, despite the fact that their data is outdated, having been collected in 1993, it focused on the effects of an intervention. Another study [15] examined the beliefs of Brazilian runners who named the choice of running shoes, training load and exercise methods (e.g., not stretching) as the main risk factors. Yet, with 95 respondents, their sample was rather small for a population-based study. In view of the limitations of the available studies, the present survey aimed to more precisely delineate the prevalence of pain as well as the personal characteristics and attitudes of runners participating in a mass event.

## 2. Materials and Methods 

### 2.1. Study Design and Ethical Considerations

An electronic web-based cross-sectional survey was performed in June 2019, immediately prior to the JP Morgan Corporate Challenge running event in Frankfurt/Main, Germany. This study followed the recommendations for *Good Practice in the Conduct and Reporting of Survey Research* [16]. Ethical approval was obtained from the local committee (Ethics Committee of the Faculty of Psychology and Sports Sciences at Goethe University Frankfurt, Germany) and all the participants gave digital informed consent. After being provided with information on aims, content, anonymous data collection and data processing on the first page of the questionnaire, they were explicitly advised that by clicking the “continue” button, they would agree to participate in the study, accepting the aforementioned terms and conditions. 

### 2.2. Sample

The JP Morgan Corporate Challenge is a series of running events organized in several cities of Northern America, Europe, Australia and Asia. With 63,000 participants, the event in Frankfurt/Main is one of the largest road races of the world. The participants, covering a distance of 5.6 kilometers (3.5 miles), are organized in teams and must be registered by their employing company.

The weather conditions at the day of the race were good, with 20 °C Celsius and the absence of rain. The target population of our study comprised participants of both sexes aged 18 years and older. Except for the participation in the upcoming race, only the ability and consent to answer the questions of the survey were required for study inclusion. Six specifically trained investigators recruited potential participants by word of mouth. The questionnaire (see below) was administered digitally via a smartphone application (Typeform, Barcelona, Spain).

### 2.3. Questionnaire

The questionnaire was created using the approach of a previous study [17]. After a thorough literature review, a group consensus process was initiated. To agree on the general scope and contents of the survey, all the authors met and gathered ideas. Following the meeting, one investigator (JW) created an initial version of the questionnaire, which was then edited according to a discussion about the other authors’ feedback. Subsequently, for face validation, the questionnaire was sent to two exercise professionals who were not familiar with the project. After once again adapting the items using their feedback, the questionnaire was administered a) to two recreational runners (one male, one female) without any scientific background. 

The two runners performing the pilot testing reported good comprehensibility and preciseness of the questionnaire, making minor suggestions for improvement and clarification of some items. The final instrument is displayed in Table 1. Its first part assessed general person-related data (e.g., age, sex, daily physical activity) as well as motivations for participating in the event, while the second part referred to manifest pain and the upcoming run (e.g., target time, strategies to prevent injury). 

### 2.4. Data Processing and Statistics

Descriptive and inferential statistics were used to analyze the obtained data, as appropriate. Interval/quasi-interval scaled data (e.g., age or pain intensity) and dichotomous data (e.g., presence of pain) were reported as absolute (n) and relative (%) values, respectively. To detect systematic associations between the assessed variables, Pearson correlations (interval scaled variables), point biserial correlations (binary and (quasi)-interval scaled variables) and contingency tables with Chi-squared tests (dichotomous variables) were calculated. In case of violations to the normalcy assumption, Kendall’s tau correlations were used. 

For data analysis, the software BiAS Statistics 11.10 (Goethe University, Frankfurt/Main, Germany) was used for all analyses and significant associations among study variables were inferred at α = 0.05.

## 3. Results

### 3.1. Participant Characteristics and Profile

Out of 806 contacted runners, *n* = 720 completed the survey. Thus. The response rate was 89.3%. The average age was 33.2 ± 0.3 years with a range spanning from 18 to 68 years (Figure 1). At 55% (*n* = 396), a slight majority of the participants were men. 

One quarter of the participants (*n* = 182) were novices as the race was their first running event. According to the estimated personal target time, only eight individuals (1.1%) seemed to b professional or semi-professional runners, indicating a run in less than 20 min. The largest part (46.5%) expected a duration of 30 to 40 min and, based on these self-reported data, was probably rather untrained/unambitious. Self-reported physical activity levels were low to moderate: 8 of 10 runners stated accumulating less than 10,000 steps per day and one quarter of the entire sample did not exceed the 5000-steps-threshold. 

### 3.2. Motivation

Two dominant factors were stated most often as the reason for participation: social interaction/team building with their colleagues (76.4%) and experiencing the atmosphere of the run (50.6%). In contrast, other aspects, such as mastering a personal challenge, beating their own time record or doing something positive for their individual health status were reported less frequently (Figure 2). Surprisingly, more than one in ten persons declared that they were participating because colleagues or employees asked them to run. All motivations were reported equally often by the two sexes (*p* > 0.05). However, there was one exception: a larger share of women (24.1%) selected mastering a challenge than men did (15.5%, Chi² = 4.4, *p* = 0.03). Higher age was correlated with doing something for personal health (*r* = 0.16, 95% CI: 0.09 to 0.23, *p* < 0.001), being in contact with colleagues (*r* = 0.14, 95%CI: 0.06 to 0.21, *p* < 0.001) and experiencing the atmosphere of the race (*r* = 0.1, 95%CI: 0.02 to 0.17, *p* = 0.01) while the other motivations were not influenced by age (*p* > 0.05).

### 3.3. Beliefs Towards Injury Prevention

Stretching (75.4%) and wearing appropriate shoes (68%) were by far the most often named strategies thought to prevent musculoskeletal injuries (Figure 3). In contrast, less than one third of the runners believed that resistance and balance training would be effective in this regard. The use of insoles (18.8%) and massage or foam rolling techniques (18.5%) was still named by a substantial share, while orthoses was selected by only 29 (4%) participants. The participants’ opinions regarding the respective preventive measures were not associated with age except for a positive correlation of the belief that appropriate shoes are effective in preventing injuries (*r* = 0.16, 95%CI: 0.08 to 0.23, *p* < 0.001). Men considered resistance training (33.6% vs. 24.7%, Chi² = 6.77, *p* = 0.01) and balance training (32.1 vs. 23.8%, Chi² = 6.05, *p* = 0.01) more effective than women. The other beliefs were not different as a function of sex (*p* > 0.05).

### 3.4. Pain

A total of 158 participants (21.9%) reported pain prior to the start of the event, 24 of them in more than one location. At rest, the mean intensity was 3.2 ± 1.7/10 on a numerical rating scale, increasing to 4.7 ± 2.1/10 during activity (Figure 4). Both at rest (*n* = 1) and during activity (*n* = 4) maximum values were 9, representing very strong discomfort. Pain at rest was clinically relevant (NRS ≥3) in 38.9% of the participants with discomfort. During activity, this value increased to 72.1%.

According to the survey, the knee represented the mostly affected location (40.5% of runners with pain), followed by the lower back (26.6%), foot (22.8%) and hip (17.1%). The presence of pain was not associated with sex or age (*p* > 0.05, Table 2). However, when distinguishing between locations, there was a small association between higher age and hip pain (*r* = 0.09, 95% CI: 0.02 to 0.16, *p* = 0.01). In addition, the presence of pain before the race correlated weakly with daily physical activity (Kendall’s tau = 0.06, *p* = 0.01).

With regard to pain intensity, women reported significantly higher values at rest (3.5 ± 1.9) than men (2.8 ± 1.6; *r* = 0.2, 95%CI: 0.04 to 0.34, *p* = 0.01), but there was no difference during activity. No association between pain intensity and sex was detected (*p* > 0.05).

## 4. Discussion

To the best of our knowledge, the present study is one of the first large-scale surveys assessing pain prevalence, injury-preventive beliefs, as well as the personal characteristics of recreational runners registered for a public mass event. According to the results, the ‘typical’ participant may be described as being in early adulthood, low to moderately active, not very ambitious regarding targeted race time and taking part mainly out of social reasons.

Although studies have prospectively investigated pain in recreational runners [18,19], few have surveyed recreational runners immediately prior to starting a race. Our finding of a pre-start pain prevalence of about 21.9% is identical to the findings of an older investigation [13] which reported pain in 22% of Brazilian recreational runners participating in different mass events. While the knee was the mostly affected location in both the previous and our study, the frequency of back pain was considerably higher in the present sample. The high prevalence of pain prior to the start of the event was not the only alarming finding. A high percentage (more than the half at rest and almost three quarters during activity) of the symptoms reported was clinically relevant when considering the threshold of ≥3 on the numerical rating scale. Beyond this, five participants even reported pain in four different locations.

Stretching and the use of appropriate shoes were reported as primary strategies to prevent injury. The results align with the work of Hespanhol et al. [15], whose participants rated not stretching as the main risk factor. However, there is a remarkable contrast: while only about one third of the previous study’s participants cited stretching, it was named by around 75% of the runners in our sample. This is highly relevant because strong evidence suggests that stretching does not help to prevent injury [20,21]. Moreover, regarding the factors, the knowledge/belief of the runners is not supported by published evidence. Foam rolling and massage were chosen by one in five respondents although, to date, no study has examined its injury preventive value. Appropriate footwear, mentioned second most often, could, in fact, be effective in reducing injury prevalence. However, achieving the assumed protective effect is more complicated than often propagated. Available literature reviews [22,23] demonstrate that commonly advertised features, such as cushioning and pronation support, may not effectively reduce injury risk. Instead, subjective comfort seems to decrease the odds of sustaining running disorders [23]. The frequent choice of the strategy “wearing adequate shoes” could hence only be classified as evidence-based if runners would only rely on the criterion of perceived comfort.

Our findings may have useful implications for public health advocates, coaches and organizers of public running events. The substantial presence of (clinically relevant) pain, the rather low levels of physical activity and the lacking knowledge about the (in)-effectiveness of popular prevention strategies should make the abovenamed actuators aware that participating in a public sports event may have meaningful downsides. While non-recurring, acute pain symptoms are rarely problematic in the short-term, they could become chronic or worsen if remaining untreated and if activity levels/training volumes are increased too quickly. Therefore, although the potential benefits of initiating/increasing regular PA most likely outweigh the negative effects, they seem only fully exploitable if the participants receive appropriate support. As such, event organizers may offer focused preparation programs or best-practice flyers. Exercise professionals could, besides being engaged in this process, particularly help to tailor optimally dosed activity programs if participants have the enthusiasm to increase their physical activity levels.

Our study is not without limitations. Our survey was performed directly before the race, which allowed for rarely performed pre-activity assessment of pain. However, due to this, limited time was available, which did not allow more detailed interviews regarding specific diagnoses or used training regimes. Therefore, it would be intriguing to combine assessments some time before as well as immediately prior to the start of a sports event. Moreover, it would have been interesting to perform an additional survey post-race. This would have allowed for analysis of pain and its handling (e.g., by means of analgesics) during the event.

## 5. Conclusions

Public running events attract numerous participants with considerable shares of novices. They may therefore effectively support the increase of global physical activity levels in the general population. However, as musculoskeletal pain is prevalent and often clinically relevant and as evidence-based knowledge about preventive strategies is poor, organizers and coaches may consider offering structured preparation programs as well as tailored running-related health education.

## Figures and Tables

**Figure 1 ijerph-16-03766-f001:**
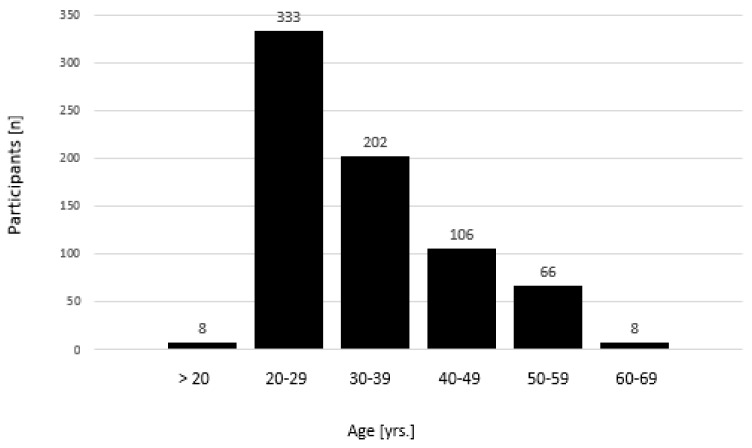
Age of the race participants in decades. Note the dominance of individuals in their twenties. Yrs. = years.

**Figure 2 ijerph-16-03766-f002:**
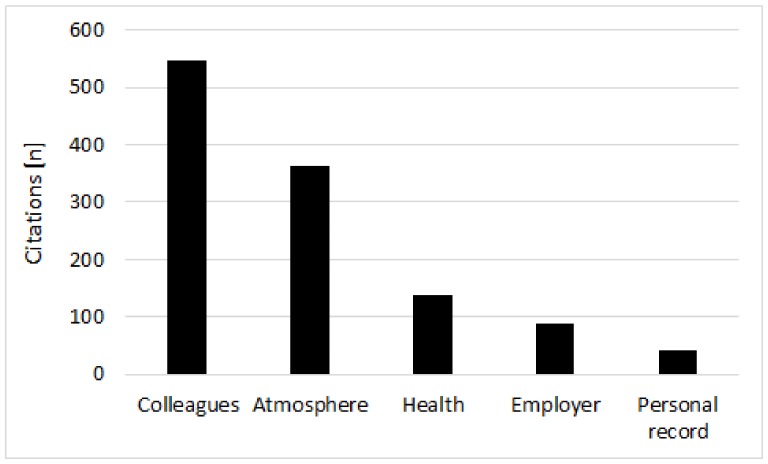
Motivations for participation as reported by the race participants.

**Figure 3 ijerph-16-03766-f003:**
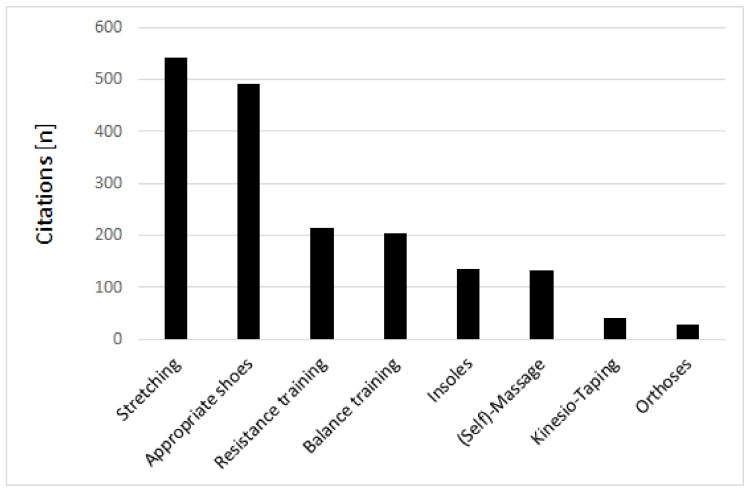
Injury prevention strategies in running as reported by the race participants.

**Figure 4 ijerph-16-03766-f004:**
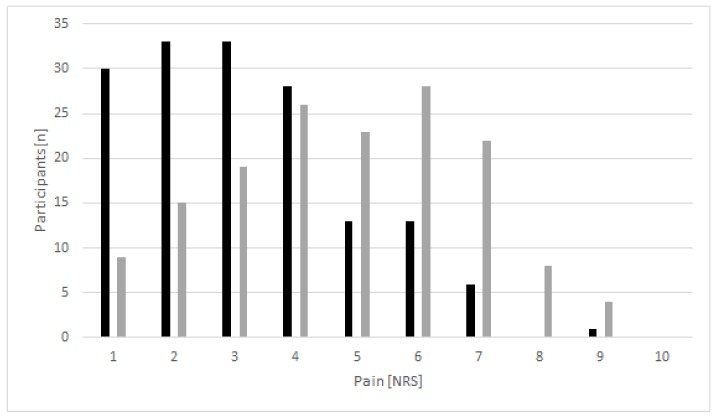
Distribution of pain intensity in race participants reporting discomfort (*n* = 158) at rest (black bars) and during activity (grey bars).

**Table 1 ijerph-16-03766-t001:** Contents of the questionnaire administered to the event participants.

Question	Answer Format
Are you male or female?	single choice
How old are you?	free entry
What is you estimated daily step count? <5.000 steps 5.000 to 10.000 steps >10.000 steps	single choice
Have you participated in an organized running event before?	yes/no
Why do you participate in the race? I want to do something positive for my personal health Contact with colleagues/team building I want to master a challenge I want to beat my personal record Because of the special atmosphere My employee or others wanted me to participate Others	multiple choice + free entry
Did you train for the race?	yes/no
What is your personal target time? <20 min 20 to 30 min 30 to 40 min >40 min	single choice
Which strategies do you consider effective in preventing running injuries? Stretching Resistance training Balance trainng Orthoses Kinesio taping Appropriate shoes Insoles Massage or foam rolling others	multiple choice + free entry
Do you have pain right now?	yes/no
If yes:	free entry
How strong is your pain on a scale from 0 (on pain at all) to 10 (worst imaginable pain) at rest?	Likert scale (10 items)
How strong is your pain on a scale from 0 (on pain at all) to 10 (worst imaginable pain) during movement?	Likert scale (10 items)
Where do you have pain? Foot Knee Hip Back Other location	multiple choice + free entry

**Table 2 ijerph-16-03766-t002:** Pain, stratified by sex and location reported by the race participants.

Location	Males	Females	Total
Foot	18	18	36
Knee	36	28	64
Hip	11	16	27
Lower Back	18	24	42
Others	15	15	30
Total	98 *	101 *	199 *

* 41 participants reported multiple pain locations. Therefore, the numbers presented higher than the absolute number of runners reporting pain (*n* = 158).

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
