# Peer review of "Why Are You Running and Does It Hurt? Pain, Motivations and Beliefs about Injury Prevention among Participants of a Large-Scale Public Running Event"

_ijerph, 2019, doi:10.3390/ijerph16193766_

Round 1

Reviewer 1 Report

General comments:

The paper is quite interesting, well written and organized. However, there are several drawbacks of the paper which must be improved. Authors should add information about the number of people at a certain age or age ranges, about weather during the run, especially the temperature. Please provide the reliability of pain scale. Did runners from many countries participate in the study, and if so, did this affect the results of the study. Please also compare the results to longer runs such as a half marathon.

Specific comments:

Line 19 I’m not sure is that study elucidate the prevalence of musculoskeletal pain or is it evaluate level of knowledge about injury prevention among recreational runners?

Line 24 - please specify exactly how many people are quarter

Line 27- please add the exact amount in brackets (1/5) or as a percentage

Line 28-29: „..3.2 on a numerical rating scale” I suggest to add range of the scale, for e.g. 3.2/10.

Line 40 – I’m not sure that this shortcut is needed, author used PA only three times in paper.

Line 42-43:  „Notwithstanding, although there is a large body of evidence supporting the health beneficial effects of PA..” Need reference(s) to support this statement.

Line 45 [3] Are you sure this position is correct? I found in page 380:  List of common medications.

Line 47-49 this sentence is not precise.  “These alternate scenarios resulted in >533,000 and >1.3 million deaths potentially avoided worldwide each year.” “new estimate of years gained worldwide increased to a median of 0.92 (range, 0.49–1.25) years”.

Line 57 – please add after 2012 yr.

Line 63 – please add more information about Brazilian runners, how many person, what distance

Line 63: Please clarify and highlight why examination prior to starting a race is distinct from others, how it can affect results?

Line 60-69: the authors wrote “Despite the running boom in western countries, the prevalence of pain during mass events as well as the‘typical participant of public mass events, his motivations to participate as well as his knowledge towards effective injury prevention are largely unknown”. However there are available research on this subject:

Injury prevention education: van  Mechelen W, Hlobil H, Kemper HC, Voorn WJ, de Jongh HR. 1993; 21(5):711-9. Prevention of running injuries by warm-up, cool-down, and stretching exercises.

Lopes AD1, Costa LO, Saragiotto BT, Yamato TP, Adami F, Verhagen E. “What is the prevalence and nature of musculoskeletal pain in recreational runners immediately before a race?”  J Physiother. 2011;57(3):179-82.  Musculoskeletal pain is prevalent among recreational runners who are about to compete: an observational study of 1049 runners.

Please add information based on this research

Line 86 - on the race's website you can find other information: “The annually and extremely popular corporate run “J.P. Morgan Challenge” took place on 12th of June 2019. Approximately 63.000 people in Frankfurt participated in that and ran in total a 3.5 mile (5.6km) race series for a charitable cause”

Line 122: probably no letter  “i” in Statstics

Line 126 – 129 - I am asking for more information related to age, or charting the number of people depending on age and gender. Did the authors considered using different stimulants or painkillers by runners? For example caffeine reduce the pain during the exercise. If not, please refer to this in the discussion or give it as a limitation of the study.

Line 162 – Older , younger – please be more accurate

Line 198 – 199 no space between the lines needed

Line 208 – change (e.g. [17,18] to [17,18]

Line 208-209: Which studies? I suggest to add references. Also as I mentioned before, why examination prior to starting a race is distinct from others? how it can affect results? can results be compared with others? Please clarify

Line 210 and 211 change percent to %

Line 222- : ? or it should by “.” Or small letter

Line 217: Expand the abbreviation

Line 226-228: Foam rolling and massage have one or more published meta-analyses and reviews on the topic that should be discussed and cited.

Line 243-244: Are these sentences the author's opinions? I suggest to add reference(s) to support this statement.

The bibliography should be improved to be consistent, as per the example:

Bowman, C.M.; Landee, F.A.; Reslock, M.A. Chemically Oriented Storage and Retrieval System. 1. Storage and Verification of Structural Information. J. Chem. Doc. 19677, 43-47; DOI:10.1021/c160024a013.

Line 272: Evironment change to shortcut Environ

Line 280: 380(9838:21- 280-27 to 380(9838):219-29.

Line 286: should be  2011; 33(3)

Line 289: remove a space 15(2):148-159.

Line 294: 38 to 38(5)

Line 305: It should be like NCBI: Lopes, A.D., Costa, L.O., Saragiotto, B.T., Yamato, T.P., Adami, F., Verhagen, E.

Line 308: 57 to 57(3)

Author Response

Reviewer 1

The paper is quite interesting, well written and organized. However, there are several drawbacks of the paper which must be improved. Authors should add information about the number of people at a certain age or age ranges, about weather during the run, especially the temperature. Please provide the reliability of pain scale. Did runners from many countries participate in the study, and if so, did this affect the results of the study. Please also compare the results to longer runs such as a half marathon.

A figure displaying the age of the participants in decades has been added. Details about weather including temperature have been added Regarding reliability: This has not been analyzed because there were no repeated measures (test-retest reliability) and because the survey was digital and not dependent on an investigator (interrater reliability). We have no information on nationality of the participants. There is only one study on pre-race pain, which had a similar distance. This has now been indicated in the manuscript.

Line 19 I’m not sure is that study elucidate the prevalence of musculoskeletal pain or is it evaluate level of knowledge about injury prevention among recreational runners?

Evaluating the prevalence of pain was a primary aim, however, we added the aspect of knowledge about injury prevention to be concise here.

Line 45 [3] Are you sure this position is correct? I found in page 380:  List of common medications.

Thank you, we agree this was a bit unclear. Recommendations are provided in different sections and thus we cited the entire book, the 380 p referred to the amount of pages of the book. This has now been removed for clarification.

Line 47-49 this sentence is not precise.  “These alternate scenarios resulted in >533,000 and >1.3 million deaths potentially avoided worldwide each year.” “new estimate of years gained worldwide increased to a median of 0.92 (range, 0.49–1.25) years”.

We have corrected the numbers (please kindy not that your first citation applies to reduction but not elimination of inactivity).

Line 63 – please add more information about Brazilian runners, how many person, what distance

Thank, you, more information have been added accordingly.

Line 63: Please clarify and highlight why examination prior to starting a race is distinct from others, how it can affect results?

Thank you very much, we have now described the relevance of pre-race pain more clearly. While general prospective registration is important for judging injury risk, pre-race pain can have severe acute consequences like altered biomechanics through compensation, worsening of structural damage (if present) or even affect the psycho-physical linkage between a person and its activity behavior. It is therefore substantially different to assessment of pain over a longer time frame, which additionally does not or only rarely tells if persons actually run/compete in pain or not.

Line 60-69: the authors wrote “Despite the running boom in western countries, the prevalence of pain during mass events as well as the‘typical participant of public mass events, his motivations to participate as well as his knowledge towards effective injury prevention are largely unknown”. However there are available research on this subject:

Injury prevention education: van  Mechelen W, Hlobil H, Kemper HC, Voorn WJ, de Jongh HR. 1993; 21(5):711-9. Prevention of running injuries by warm-up, cool-down, and stretching exercises.

Lopes AD1, Costa LO, Saragiotto BT, Yamato TP, Adami F, Verhagen E. “What is the prevalence and nature of musculoskeletal pain in recreational runners immediately before a race?”  J Physiother. 2011;57(3):179-82.  Musculoskeletal pain is prevalent among recreational runners who are about to compete: an observational study of 1049 runners.

Please add information based on this research

We have expanded the description of the available literature, clearly specifying the research deficit.

Line 86 - on the race's website you can find other information: “The annually and extremely popular corporate run “J.P. Morgan Challenge” took place on 12th of June 2019. Approximately 63.000 people in Frankfurt participated in that and ran in total a 3.5 mile (5.6km) race series for a charitable cause”

Thank you, we have corrected these numbers.

Line 126 – 129 - I am asking for more information related to age, or charting the number of people depending on age and gender. Did the authors considered using different stimulants or painkillers by runners? For example caffeine reduce the pain during the exercise. If not, please refer to this in the discussion or give it as a limitation of the study.

Thank you, we agree this would have been highly interesting. As explained in the limitations section, there was no time to ask for more. Regarding caffeine, please note that we did pre-race interviews and hence, an intervention or data collection post-race was not possible. However, we have added some of the examples given here in the corresponding part of the discussion.

Line 162 – Older , younger – please be more accurate

Thank you, this may indeed have been misleading. There was no correlation of age and the beliefs. We have corrected this phrase accordingly.

Line 198 – 199 no space between the lines needed

Must have been inserted by editorial office, looks nice in submitted version.

Line 208-209: Which studies? I suggest to add references. Also as I mentioned before, why examination prior to starting a race is distinct from others? how it can affect results? can results be compared with others? Please clarify

Thank you, as per your previous comment, we have now clarified this. In addition, we have now made explicit that available study were mostly prospective.

Line 226-228: Foam rolling and massage have one or more published meta-analyses and reviews on the topic that should be discussed and cited.

Please kindly note that these do not provide data on injury prevention. For instance, regarding foam rolling, data is only available regarding ROM improvements. We therefore consider out position that the injury preventive value is, at best, unclear, valid.

Line 243-244: Are these sentences the author's opinions? I suggest to add reference(s) to support this statement.

Reviewer 2 Report

P3 L73 immdiately..............  change to inmediatly

P5-L134  How did they determine the degree of training of the participants?

P7-L185 . How many %?

P7-L190    I do not understand :  With regard to pain intensity, women reported significantly higher values at rest (3.5±1.9) than men (2.8±1.6; r=0.2, 95%CI: 0.04 to 0.34, p = 0.01),

In figures 1 and 2 represent SD or EE

Do you study the Reliability of questionnaire?

NRS? What does it mean?

Do you think such weak correlations are relevant to the st

Author Response

P5-L134  How did they determine the degree of training of the participants?

We assumed this based on the reported expected running times. We have now made this more explicit.

P7-L185 . How many %?

Percent values have been added.

P7-L190    I do not understand :  With regard to pain intensity, women reported significantly higher values at rest (3.5±1.9) than men (2.8±1.6; r=0.2, 95%CI: 0.04 to 0.34, p = 0.01),

Can you specify? The sentence looks easy-to-understand to us.

In figures 1 and 2 represent SD or EE

Please kindly refer to the labels, the figures show total mentions (citations [n]).

Do you study the Reliability of questionnaire?

No reliability has not been analyzed because there were no repeated measures (test-retest reliability) and because the survey was digital and not dependent on an investigator (interrater reliability).

NRS? What does it mean?

Numerical rating scale, this has been clarified.

Do you think such weak correlations are relevant to the st

We agree there were no strong correlations, most can be classified as weak to moderate. However, we think they are appropriately reported and discussed as we do not speculate based on them or provide tendencies for overestimation.